# The Early Exposure Rate and Vertical Bone Gain of Titanium Mesh for Maxillary Bone Regeneration: A Systematic Review and Meta-Analysis

**DOI:** 10.3390/dj13020052

**Published:** 2025-01-23

**Authors:** Felice Lorusso, Sergio Alexandre Gehrke, Iris Alla, Sergio Rexhep Tari, Antonio Scarano

**Affiliations:** 1Department of Innovative Technologies in Medicine & Dentistry, University of Chieti-Pescara, 66100 Chieti, Italy; felice.lorusso@unich.it (F.L.); iris.alla@unich.it (I.A.); rexhep.tari@studenti.unich.it (S.R.T.); 2Department of Research, Bioface/PgO/UCAM, Montevideo 11100, Uruguay; sergio.gehrke@hotmail.com

**Keywords:** bone regeneration, titanium mesh, jaws defects

## Abstract

**Background/Objectives:** The use of titanium meshes in bone regeneration is a clinical procedure that regenerates bone defects by ensuring graft stability and biocompatibility. The aim of the present investigation was to evaluate the clinical effectiveness of titanium mesh procedures in terms of vertical bone gain and the exposure rate. **Methods:** The product screening and eligibility analysis were performed using the Pubmed/MEDLINE, EMBASE, and Google Scholar electronic databases by two authors. The selected articles were classified based on the study design, regenerative technique, tested groups and materials, sample size, clinical findings, and follow-up. A risk of bias calculation was conducted on the selected randomized controlled trials (RCTs) and non-randomized trials and a series of pairwise meta-analysis calculations were performed for the vertical bone gain (VBG) and exposure rate. A significantly lower exposure rate was observed using coronally advanced lingual flaps (*p* < 0.05). No difference was observed between the titanium mesh and GBR techniques in terms of VBG (*p* > 0.05). **Results:** The initial search output 288 articles, and 164 papers were excluded after the eligibility analysis. The descriptive synthesis considered a total of 97 papers and 6 articles were considered for the pairwise comparison. **Conclusions:** Within the limits of the present investigation, the titanium mesh procedure reported high VBG values after the healing period. The mesh exposure rate was drastically lower with passive management of the surgical flap.

## 1. Introduction

The treatment of severe bone ridge atrophies represents a complex clinical challenge in oral surgery due to the dysmorphic alteration of the oral tissues and the loss of support for implant rehabilitation [1,2,3]. Alveolar bone defects are commonly due to the loss of teeth, and from dentofacial traumas, neoplasms, and cyst expansion and removal defects, while the resorption rate could be severely increased other factors including infections and passive loading by an incongruous prosthesis [4,5]. On the other hand, different resorption patterns have been described between the horizontal and vertical components of the bone ridges of both the mandibular and maxillary ridges [2,4,5,6,7]. For this purpose, several different bone augmentation procedures for increasing the bone volume have been purposed including inlay/onlay bone grafts, bone distraction, guided bone regeneration, and titanium meshes [8,9,10,11]. Guided bone augmentation procedures are accomplished using the creation of a regenerative space based on scaffold positioning to stabilize the blood clot [12]. The addition of a covering made of collagen membranes has been used to compartmentalize the oral tissue components to restore the oral tissues’ anatomical morphology [6,13]. In the literature, this is described as the application of a non-resorbable membrane (i.e., polytetrafluoroethylene (PTFE)) or resorbable membrane (i.e., collagen) [14,15,16]. Historically, non-resorbable and titanium-reinforced membranes were used for guided bone regeneration procedures in the late 1980s due to their high mechanical stability and ability to maintain spaces [17]. Limitations of this technique include the necessity for a second surgery to remove the mesh and the tendency for exposure during the healing period [17]. Titanium meshes are a space-making device that have been used for the treatment of complex vertical defects due to the addition of a bone graft [11]. The theoretical advantage of titanium meshes is the presence of pores, which are able to create a favorable environment for the vascular sustenance and integration of the core graft [11,18]. Another interesting characteristic is the documented high biocompatibility of titanium, which prevents foreign body reactions and reduces the failure rate of the procedure [11,19]. Mechanically, the mesh is characterized by a high ductility due to the adaptation of the device to the bone defect area [11,19]. In addition, the device rigidity is able to guarantee a regenerative space during the healing phase and the graft integration process [11,19]. In the literature, the customization of titanium meshes through CAD/CAM has been used to increase the stability of regenerative devices and the on-chair procedure duration [20,21,22]. However, titanium meshes are technically sensitive and they are not free from complications. In fact, the main source of titanium mesh failure is from exposure of the mesh during the healing period, combined with contamination of the bone graft, which often irreversibly compromises the regenerative procedure [23]. The aim of the present systematic review was to investigate the clinical effectiveness of titanium meshes in bone regeneration procedures.

## 2. Materials and Methods

### 2.1. Screening of Scientific Articles

The literature search was performed following the criteria of the PICO guidelines (population, intervention, comparison, and outcome), as shown in Table 1. The data collected from the systematic search were processed in accordance with the Preferred Reporting Items for Systematic Reviews and Meta-Analyses (PRISMA) guidelines. The present review was registered in the PROSPERO database (CRD42024585970). The Boolean search was carried out according to the strategy described in Table 2 and performed on the PubMed, EMBASE, and Google Scholar electronic databases (10 June 2024).

### 2.2. Inclusion and Exclusion Criteria

In the initial screening phase, the identified studies were assessed based on the following inclusion criteria: human clinical trials, prospective or retrospective studies, case series or case reports, with no restriction on follow-up after surgical procedures or regarding the type of graft material mix used, and finally English-language papers.

The exclusion criteria were systematic literature reviews, editorial letters, in vitro studies, and animal studies. The evaluation of the manuscripts using the above criteria was performed for the purpose of including them in the eligibility analysis.

### 2.3. Screening Process

Two expert reviewers (FL and IA) independently and blindly performed the selection and screening of articles in order to identify scientific articles for the analysis processes. However, the articles that were excluded from the research work according to the criteria are reported in the paper as well as the justification for their exclusion.

### 2.4. Data Analysis

A database was specifically created using Excel software (Microsoft, Redmond, WA, USA) to enter the data collected from the included scientific studies. The collected data were classified according to the following characteristics: study design, sample size, regenerative technique, complications, biomaterial/resorbable membrane type, surgical flap technique, and follow-up.

### 2.5. Outcome Measures

The outcome measures considered for the data analysis were the occurrence of flap exposure during the bone regeneration healing period (<6 months), and the vertical bone height and horizontal bone gains calculated at the follow-up using computed tomography assessments.

### 2.6. Risk of Bias Assessment (RoB)

An RoB analysis was performed according to the OHAT Guidelines and Risk of Bias Rating Tool for Human and Animal Studies using Rev Man 5.5 (The Nordic Cochrane Centre, The Cochrane Collaboration, Copenhagen, 2014). Only the trials included for the meta-analysis process were submitted to the risk of bias assessment [1,2,3,4,5,6,7,8,9,10,11,12,13,14,15,16,17,18,19,20,21,22,23,24,25,26,27,28,29,30]. The following guideline criteria were applied: randomization sequence, allocation concealment, blinding participants, blinding outcomes, incomplete outcome data, selective reporting, and other biases. The RoB parameters were classified as adequate, unclear, or inadequate. The minimum RoB ratio was a total of 5/7 low risk (lr) indicators with/without unclear risk (ur) parameters. Otherwise, the articles were categorized as high risk (hr).

### 2.7. Meta-Analysis

A forest plot of the relative effects was generated to assess the consistency and the significance of the rankings. I^2^ < 40% was considered low heterogeneity. To guarantee valid pairwise comparisons, we selected studies with similar methodologies for further statistical calculations. Pairwise comparisons were performed for the titanium mesh group vs. membrane regeneration group and the coronally advanced lingual flap group vs. control group considering the site exposure and vertical bone gain (VBG) parameters. The exposure rate was expressed as an Odds Ratio (OR) and the VBG was expressed as the mean difference (MD).

## 3. Results

### 3.1. Screening Procedure

The search conducted using electronic databases (PubMed/Medline, EMBASE, and Google Scholar) found 288 articles. The search did not detect duplicates so the scientific articles were submitted for eligibility evaluation. A total of 164 articles were excluded from the synthesis process for reasons such as being off-topic (114 articles), being in a different language (15 publications), using an animal model (41 scientific articles), and being a literature review (21 papers). As a result of the careful selection, a total of 97 scientific articles were included in the descriptive analysis and 6 articles were considered for the pairwise meta-analysis. This systematic literature review included retrospective case–control studies, prospective studies, cohort studies, case series and case reports, randomized controlled trials, non-RCTs, preliminary studies, and comparative studies (Figure 1; Table 2).

### 3.2. Characteristics of the Included Studies

The descriptive synthesis reported that the most frequent grafts used for bone regeneration were autogenous bone and autogenous bone mixed with a heterologous bone graft. Some studies differed in the autogenous/heterologous mix ratio, which ranged from a ratio of 50:50 to 70:30 of autogenous bone and BBM (bovine bone mineral). The combination of platelet-rich plasma (PRP), collagen sponges (rhBMP-2 ÷ ACS), resorbable collagen membranes, and alloplastic materials mixed with a nano-bone graft was also reported (Table 3 and Table 4). The most frequent complication reported was mesh exposure that was correlated to a partial failure of the graft or, in some cases, a higher incidence of compromised bone grafts. Other reported complications were infection, total/partial bone resorption, temporary neurological disturbances, and implant failure. The follow-up results were heterogeneous since the follow-up time in the included studies ranged from 5 months to 20.5 years (Table 3 and Table 4).

### 3.3. RoB Findings

The summary of the RoB assessment results is presented in Figure 2. According to the Cochrane Collaboration, most of the studies were considered to have a low risk of bias. According to the selection bias assessment, the findings were 71.4%lr and 28.6%hr regarding random sequence generation. The performance bias and detection bias analyses reported 28.6%lr and 71.4%ur for these studies. The attrition bias was 28.6%ur and 71.4%lr (Figure 3). A value of 100%lr was reported for the allocation concealment, selective reporting, and other biases.

### 3.4. Meta-Analysis Assessment

#### 3.4.1. Titanium Mesh vs. Membrane GBR: Site Exposure

This assessment included four articles for a total of 130 participants (range: 20–40). The estimated effect was 2.56 [0.91; 7.20]. The heterogeneity test reported a Chi2 value of 0.91 and I2 of 0%. No significant differences between the study groups were reported in terms of exposure of the site during the healing period (*p* = 0.08) (Figure 4). The exposure ratio of the titanium mesh and GBR groups were, respectively, 21.53% and 9.23%.

#### 3.4.2. Titanium Mesh vs. Membrane GBR: Vertical Bone Gain (VBG)

This assessment included four articles for a total of 130 participants (range: 20–40). The estimated effect was −0.12 [−0.81; 0.58]. The heterogeneity test reported a Chi2 value of 6.43 and I2 of 53%. No significant differences between the study groups were reported in terms of the VBG (*p* = 0.74) (Figure 5). The mean VBG of the titanium mesh and GBR groups were, respectively, 4.22 ± 1.68 and 4.42 ± 1.18.

#### 3.4.3. Coronally Advanced Lingual Flap Site Exposure

This assessment included two articles for a total of 54 participants (range: 14–40). The estimated effect was 0.10 [0.01; 0.94]. The heterogeneity test reported a Chi2 value of 0.66 and I2 of 0%. Significant differences between the study groups were reported in terms of the exposure of the site during the healing period (*p* = 0.04) (Figure 6). The exposure ratios of the coronally advanced lingual flap and control groups were, respectively, 0% and 43.2%.

#### 3.4.4. Coronally Advanced Lingual Flap Vertical Bone Gain (VBG)

This assessment included two articles for a total of 54 participants (range: 14–40). The estimated effect was 0.91 [−0.89; 2.72]. The heterogeneity test reported a Chi2 value of 5.06 and I2 of 80%. No significant differences between the study groups were reported in terms of the VBG (*p* = 0.32) (Figure 7). The VBG of the coronally advanced lingual flap and control groups were, respectively, 3.68 ± 1.36 and 3.26 ± 1.47.

## 4. Discussion

The use of titanium mesh in the reconstruction of localized bone defects has been used with high reliability and very low exposure and complication rates [112]. Titanium mesh has been indicated for a wide range of clinical defects including peri-implant bone defects, maxillary atrophy, alveolar sockets, and periodontal defects, and for other therapeutic applications [90]. The literature documents its use in a small number of cases for more extensive defects that originate from neoplastic pathological processes, such as odontogenic keratocyst processes, and trauma treated with complete ostectomy, hemibulectomies, and completely disarticulated resections of mandible and mandibular rami [27,113]. There are documented cases of titanium mesh use in non-grafted sinus floor elevation [114]. Titanium meshes are high-performance devices with high biocompatibility; their the barrier effect can guide the healing processes in the absence of immune responses during healing [102]. The grids can be morphologically adapted to the defect which makes them highly specific and customizable; such customization can be achieved using laser sintering or CAD/CAM [102]. Titanium grids can be stabilized with microscrews on the sides of the membrane itself and can be equipped with holes that allow for a greater blood supply to the defect, bringing oxygen, nutrients, and immune cells into the defect, which are essential to ensure the success of osteogenesis. Studies have confirmed that macroporosity has effects on bone regeneration by ensuring a sufficient blood supply to the defect, stimulating osteogenesis due to the presence of the holes. It has been observed that titanium meshes do not interfere with blood flow [21]. In addition, the presence of the mesh, compared with the presence of resorbable membranes alone, ensures that the treatment is not compromised; thus, they are considered reliable for promoting new bone formation. The rigidity of the titanium mesh ensures stability and prevents the collapse of the membrane itself within the defect, a situation that is possible with the use of resorbable membranes alone [25,56,115]. In some cases of combination regenerative and implant therapies, the implants were placed concurrently with the titanium mesh; in other cases, the placement of the titanium mesh occurred in a second surgery after 8–9 months [107]. Regenerative procedures using titanium meshes resulted in significant bone regeneration in the narrow alveolar ridges, allowing for implant placement [39]. Regenerative site exposure seems to be one of the most common early and delayed complication during the healing period. The present investigation reported no significant differences in exposure rate for titanium mesh vs. membrane GBR procedures (*p* = 0.06). Fewer exposures were observed with the use of e-PTFE membranes to cover the titanium meshes [15]. Due to the tight spread of the study outcome and the limited number of selected articles, this aspect deserves more study to determine the exposure outcome. A critical point of the present investigation was to analyze the wide heterogeneity in methods including the treatment site and jaw region, defect extension, simultaneous/delayed implant positioning, graft materials, and additional screws and plates used.

Following to the Cochrane review methodology, the present review performed a search in multiple electronic databases. Due to the difficulties in finding MesH term indicators for this topic, the screening was conducted considering all clinical studies and without applying filters regarding the study design methodology for the full-text evaluation, eligibility analysis, and descriptive synthesis. The statistical methods considered the applicability of sub-group comparisons. The main limit of this approach is indubitably a decrease in the study data robustness and strength that should be considered when interpreting the findings from this review. The present study considered an observation period of only about 9 months but a more extended follow-up period is necessary to evaluate the medium- and long-term effectiveness of both techniques in over to evaluate the comparative performance of mesh regeneration compared to membrane grafts.

Our opinion is that homogeneous study methodologies are necessary to improve the robustness of meta-analyses. It is necessary for review methodologies to reduce the wide range of biases associated with several variables including the surgical technique, procedure site (single/multiple edentulism), atrophy grading, mesh characteristics (including the porosity), adaptation technique, use of stabilization screws, and biomaterial used. In fact, considering the wide range of biomaterials used and the differences in methodology, a meta-analysis was not possible.

A sufficient pool of articles for a pairwise comparison was only possible for an analysis of the VBG and coronally advanced lingual flap as statistical variables based on the methodological characteristics and RoB of the considered studies. The VBG also seemed to be similar for both clinical protocols; more histological comparisons could elucidate the graft-interface differences and the new bone formation patterns between the procedures. A drastic reduction in the exposure rates was reported for the coronally advanced lingual flap method, suggesting that it could be considered a favorable approach for decreasing the incidence of complications. No effects were reported for the vertical bone gain parameters. In addition, treating large defects with a customized titanium mesh is a useful protocol and provides a predictable result, even in the case of dehiscence. Custom, pre-formed titanium mesh together with a mixture of autologous bone and a xenograft is a feasible and reliable technique for vertical bone regeneration and advanced and three-dimensional defects [95].

## 5. Conclusions

Due to the weak robustness of the study data, the limitations of the present review, and the strength of the analytic findings, no definitive conclusions could be made but this topic is worthy of further investigation in the future. The research outcome seems to suggest that bone regeneration of more extensive defects using titanium meshes represents a useful bone regeneration technique, which, despite being performed with different methods using different combinations of membranes and/or bone grafts of different types, and its possible complications, was found to not compromise regenerative techniques. In the present investigation, no significant differences in bone exposure and vertical bone gain were observed when comparing the technique with membrane bone regeneration. The physical and morphological characteristics of the titanium meshes, which can also be customized to the conformation of the defect, guarantee the immobilization and stability of the defect and thus will guide the regeneration and, when present, the optimal integration of the biomaterial. The management and the surgical passivity of the flaps seems to minimize the risk of exposure, with a significant reduction in the complication incidence.

## Figures and Tables

**Figure 1 dentistry-13-00052-f001:**
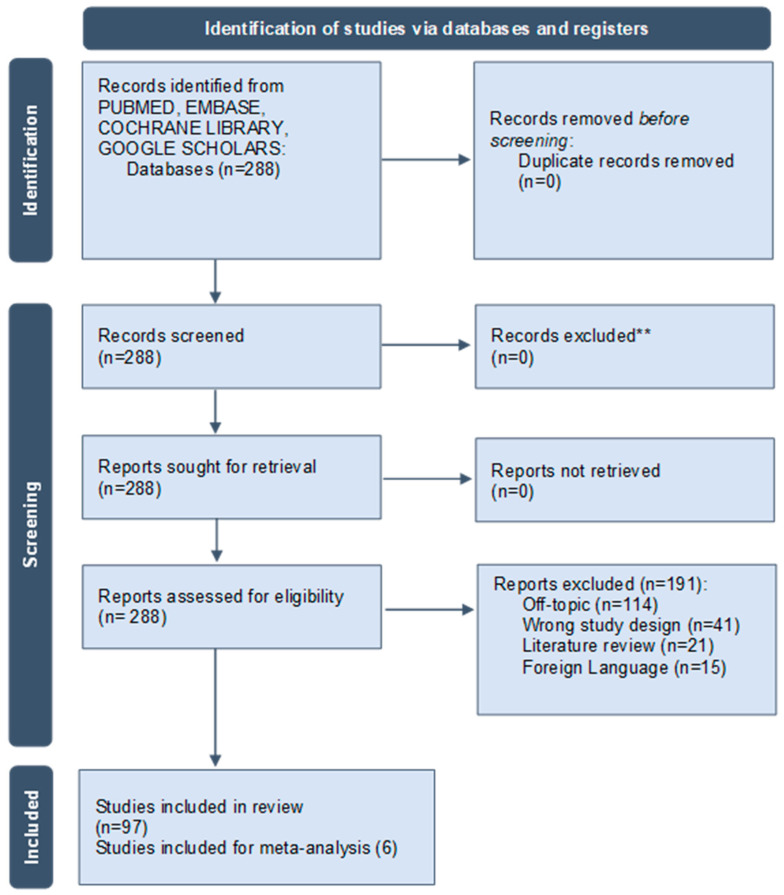
Screening flowchart for the investigated studies following the PRISMA guidelines. ** the step was performed by human with no automation tools.

**Figure 2 dentistry-13-00052-f002:**
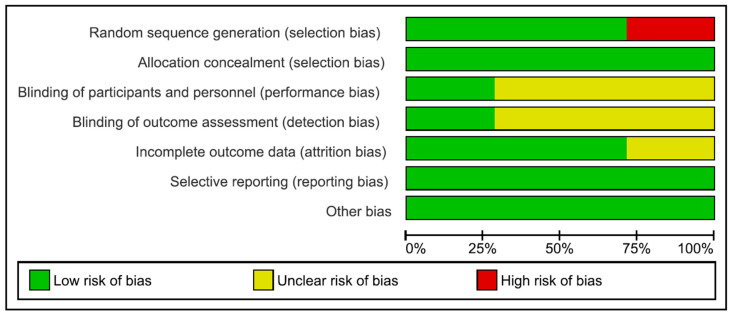
Risk of bias graph: review authors’ judgements about each risk of bias item, presented as percentages for all included studies.

**Figure 3 dentistry-13-00052-f003:**
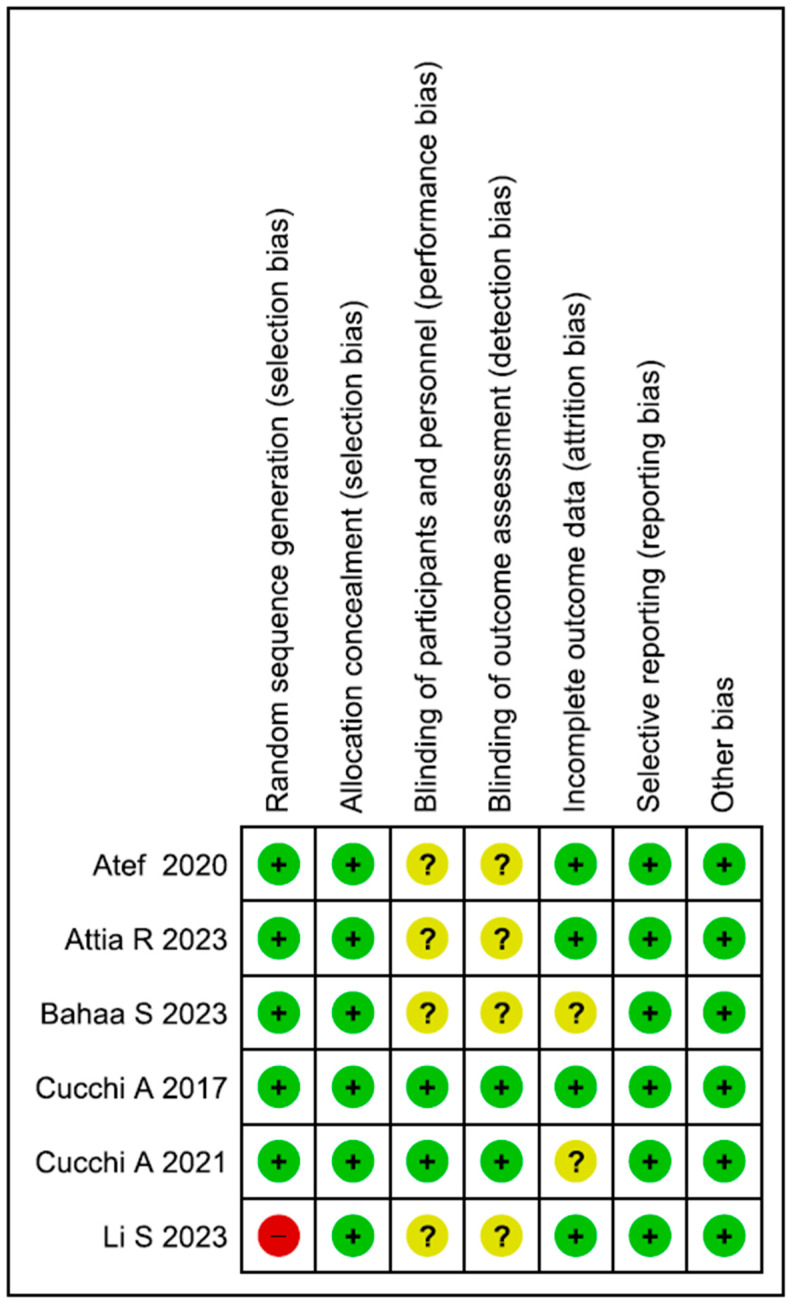
Risk of bias summary: review authors’ judgements about each risk of bias item for each included study [green light (+): low RoB; yellow light (?) uncertain RoB; red light (−): high RoB] [6,67,87,105,106,109].

**Figure 4 dentistry-13-00052-f004:**
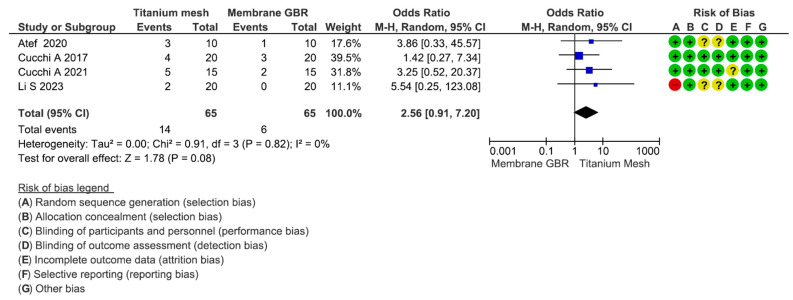
Forest plot of comparison of exposure outcome for mesh group vs. GBR group [6,67,87,109].

**Figure 5 dentistry-13-00052-f005:**
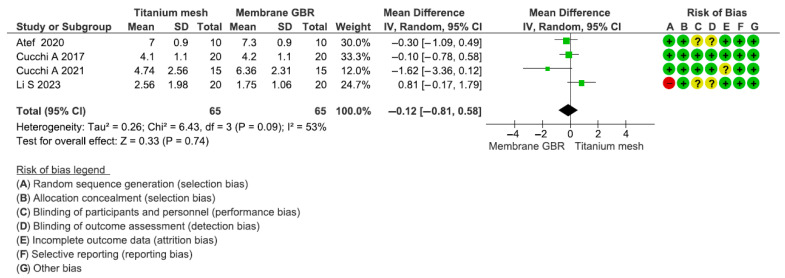
Forest plot of comparison of vertical bone gain outcome for mesh group vs. GBR group [6,67,87,109].

**Figure 6 dentistry-13-00052-f006:**
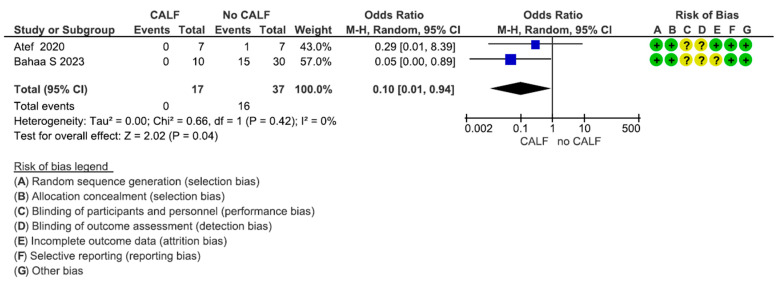
Forest plot of comparison of lingual flap release exposure [6,106].

**Figure 7 dentistry-13-00052-f007:**
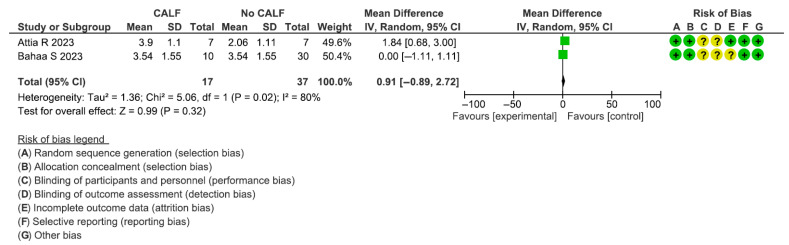
Forest plot of comparison of vertical bone gain (VBG) [105,106].

**Table 1 dentistry-13-00052-t001:** Summary of the PICO (population, intervention, comparison, and outcome) model.

Population	Intervention	Comparison	Outcomes
Subjects affected by severe bone ridge atrophy and are candidates for a graft	-Titanium mesh regeneration procedure	-Bone regeneration with a resorbable membrane	-Vertical bone gain -Mesh exposure

**Table 2 dentistry-13-00052-t002:** Screening strategy using Boolean search.

Search Strategies	
Keywords	(“titanium mesh” OR “titanium frames” OR “titanium membranes”) AND (“bone augmentation” OR “bone regeneration” OR “guided bone regeneration” OR ”bone-defect”)
Databases	PubMed/Medline, EMBASE, and Google Scholar

**Table 3 dentistry-13-00052-t003:** Studies included after the literature screening [RCT: randomized controlled trial; non-RCT: non-randomized controlled trial]. The synthesis was performed considering the regenerative methods, study model design, sample size, and test and control groups.

First Author	Journal	Year	Methods	Study Design	Sample Size	Test Group	Control Group
Boyne PJ [24]	*Head Neck Surg*	1983	Ti mesh	Case series	6 patients	6 cases after neoplastic resection	-
L Malchiodi [25]	*Int J Oral & Maxillofacial Imp*	1998	Ti mesh	Case report	25 patients	-	-
von Arx T [26]	*Int J Periodontics Restorative Dent*	1998	Ti mesh	Case report	6 patients	10 implant sites	-
von Arx T [27]	*Clin Oral Implants Res*	1999	Ti mesh/microscrews	Non-RCT	15 patients	20 implants placed in GBR sites	-
Sumi Y [28]	*Oral Surg Oral Med Oral Pathol Oral Radiol Endod*	2000	Ti mesh/bone screws	Case report	3 patients	Implant placement into 3 sites	-
Klug CN [29]	*J Oral Maxillofac Surg*	2001	Ti mesh (microscrews), distractor	Case series	10 patients	Intraoral microplate distractors for severe atrophy of edentulous molar region placed in 13 sites	-
Maiorana C [30]	*Int J Oral Maxillofac Implants*	2001	Ti mesh	Non-RCT	-	-	-
Artzi Z [31]	*Int J Oral Maxillofac Implants*	2003	Ti mesh (screws)	Case report	10 patients	10 severely resorbed sites in root(screw-type implants)	-
Degidi M [32]	*J Oral Implantol*	2003	Ti mesh	Non-RCT	18 patients		
Proussaefs P [33]	*Int J Periodontics Restorative Dent*	2003	Ti mesh	Non-RCT	7 patients	7 surgical sites treated with titanium mesh and graft	-
Roccuzzo M [34]	*Clin Oral Implants Res*	2004	Ti mesh	Case report	18 patients	Ti mesh with biomaterial fixed using titanium screws	-
Proussaefs P [35]	*J Oral Implantol*	2006	Ti mesh	Non-RCT	17 patients	Ti mesh with 50:50 autogenous graft/Bio-Oss	-
Kfir E [36]	*J Oral Implantol*	2007	Ti mesh	Case report	15 patients	Ti mesh with biomaterial	-
Longoni S [37]	*The International journal of oral & maxillofacial implants*	2007	Ti mesh	Case report	1 patient	Ti mesh with biomaterial	-
Roccuzzo M [38]	*Clin Oral Implants Res*	2007	Ti mesh	RCT	23 patients	Bone graft + Ti mesh at 12 sites	12 sites (bone graft alone)
Aikawa T [39]	*Oral Surg Oral MedOral Pathol Oral Radiol Endod*	2008	Ti mesh plate	Case report	1 patient	Defect due to keratocystic odontogenic extirpation (15 yrs before)	-
Pieri F [40]	*J Periodontol*	2008	Ti mesh	Clinical trial	16 patients	19 reconstructive procedures with delayed implant (44)	-
Corinaldesi G [41]	*Int J Oral Maxillofac Implants*	2009	Ti mesh	Retrospective study	24 patients	13 Patients: 20 implants placed using reconstructive procedure	11P: 36 implants, second surgery 8 to 9 months later
Torres J [42]	*J Clin Periodontol*	2010	Ti mesh	RCT	30 patients	15 P: Ti meshes covered with PRP	15P: Ti meshes with no PRP
Ciocca L [43]	*Med Biol Eng Comput*	2011	Customized Ti mesh Direct metal laser sintering (DMLS)	Case report	1 patient	A 53-yo M subject treated with 3D customized titanium mesh	-
Misch CM [44]	*Int J Periodontics Restorative Dent.*	2011	Ti mesh	Case report	5 patients	Ti mesh + rhBMP-2 ÷ ACS	-
Cicciù M [45]	*Open Dent J*	2012	Ti mesh plate, monocortical screws	Case report	1 patient	Defects associated with dentinogenic ghost cell tumor; Ti plate and mesh + absorbable collagen sponge + rhBMP-2	-
Her S [46]	*J Oral Maxillofac Surg*	2012	Ti mesh, titanium screws	Retrospective study	26 patients	27 sites: bone grafting + fixation of titanium mesh	-
Miyamoto I [47]	*Clin Implant Dent Relat Res*	2012	Ti mesh	-	41 patients	50 surgical sites	-
Ciocca L [48]	*Comput Methods Biomech Biomed Engin*	2013	Ti mesh, titanium screws	Case report	1 patient	Bone defect in a 46-yo male subject	-
Funato A [49]	*Int J Periodontics Restorative Dent*	2013	Ti mesh	Retrospective study	19 patients	-	-
Atef M [50]	*Clin Implant Dent Relat Res.*	2014	Ti mesh	Case series	4 patients	8 maxillary sinus sites treated with titanium mesh elevation	-
Butura CC [51]	*Int J Oral Maxillofac Implants*	2014	Ti mesh Rigid screw fixation	Case report	7 patients	23 compromised alveolar sites underwent extraction and debridement	-
Jung GU [52]	*J Korean Assoc Oral Maxillofac Surg*	2014	Ti mesh	Preliminary study	10 patients	Sites treated with biomaterial covered by Ti mesh	-
Katanec D [53]	*Coll Antropol*	2014	Ti mesh-retaining screws	Case report	61 patients	-	-
Levine RA [54]	*Compend Contin Educ Dent*	2014	Ti mesh	Case report	1 patient	Single implant in premolar region	-
Lizio G [55]	*Int J Oral Maxillofac Implants*	2014	Ti mesh	Retrospective study	12 patients	15 alveolar defects treated with Ti mesh and particulate grafts	-
Poli PP [56]	*Open Dent J*	2014	Ti mesh	Retrospective clinical study	13 patients	Ti mesh filled with intraoral biomaterial	-
Vrielinck L [57]	*J Craniofac Surg*	2014	Custom-made titanium membrane	Case Report	1 patient	Odontogenic keratocyst with remaining inferior alveolar nerve removed and curettage of the lesion; Ti plate fixed with screws	-
De Angelis N [58]	*J Periodontics Restorative Dent*	2015	Pre-adapted Ti mesh bone screws	Case series	2 patients	Surgical site: Ti mesh + rhPDGF	-
Di Stefano DA [59]	*J Contemp Dent Pract*	2015	Pre-shaped Ti mesh	Case report	1 patient	Titanium mesh graft treatment in a 54-yo patient	-
Kim Y [60]	*Dent Traumatol*	2015	Ti mesh	Case report	3 patients	Ti mesh + biomaterial + membrane	-
Lee JT [61]	*J Korean Assoc Oral Maxillofac Surg*	2015	Ti mesh	Case report	1 patient	1. Failed intra-mobile cylinder implant system 2. Failed Ti mesh 3. Distraction osteogenesis	-
Sumida T [62]	*J Craniomaxillofac Surg*	2015	Ti mesh-retaining screws	Non-RCT	26 patients	13 patients: custom-made devices	Commercial Ti mesh that was bent during operation
Misch CM [63]	*Int J Oral Maxillofac Implants*	2015	Pre-shaped Ti mesh	Retrospective study	1 patient	Titanium mesh graft treatment in 54-yo patient	-
Knöfler W [64]	*Int J Implant Dent*	2016	Ti mesh membranes as graft materials	Retrospective study	3095 patients	Titanium mesh in augmented sites	No augmented sites
Zita Gomes R [65]	*Biomed Res Int*	2016	Ti mesh for horizontal ridge defect	RCT	25 patients	40 implants with simultaneous GBR and Ti meshes	-
Ahmed M [66]	*Int J Oral Maxillofac Surg*	2017	Micro- (0.1 mm) and resorbable (0.3 mm poly-dl-lactide) Ti meshes	Case series, split-mouth study	8 patients	Bilateral sinus pneumatization sites; lateral window technique (in sinuses); elevated and maintained with resorbable membrane	Elevated and maintained with Ti meshes
Cucchi A [67]	*Clin Implant Dent Relat Res*	2017	Ti mesh	RCT	40 patients	Titanium mesh graft	PTFE-reinforced membrane
Jegham H [68]	*J Stomatol Oral Maxillofac Surg*	2017	Customized Ti mesh to shape fixing screws	Case report	1 patient	1 surgical site for implant in maxillary central incisor	-
Scarano A [69]	*Oral Implantol (Rome)*	2017	Ti mesh	Case report	3 patients	3 implant defects	-
Alagl AS [70]	*J Int Med Res*	2018	Ti mesh	Case report	1 patient	Regeneration site implant at central incisor position	-
Ciocca L [71]	*J Oral Implantol*	2018	Ti mesh (CAD-CAM-customized) -retaining screws	Non-RCT	9 patients	Implant surgery at atrophic sites	-
Inoue K [72]	*Implant dentistry*	2018	Selective laser melting titanium mesh sheet	Case series	2 patients	Laser melt titanium mesh/immediate implant	-
Lorenz J [73]	*J Oral Implantol*	2018	Ti mesh (3D planned)	Case report	1 patient	1 surgical site in a squamous cell carcinoma patient	-
Zhou M [74]	*J Oral Implantol*	2018	Ti mesh	Case report	1 patient	Bone deficiency in the No. 11 and No. 24–25 regions	-
Cucchi A [75]	*J Oral Implantol*	2019	Custom-made CAD/CAM titanium meshes	Case report	1 patient	Custom-made titanium mesh/immediate implant	-
Di Stefano DA [59]	*Dent J (Basel)*	2019	Pre-shaped Ti mesh	Case report	1 patient	GBR/implant position	-
Hartmann A [76]	*Implant Dent*	2019	Titanium mesh	Non-RCT	65 patients	Titanium mesh	-
Mounir M [77]	*Clin Implant Dent Relat Res*	2019	Ti mesh and customized poly-ether-ether-ketone (PEEK) mesh	RCT			-
Tallarico M [78]	*Materials (Basel)*	2019	Ultra-fine titanium mesh	Case series	7 patients	Ultra-fine titanium mesh/immediate implant	-
Zhang T [79]	*Clin Implant Dent Relat Res*	2019	L-shaped titanium mesh	Retrospective study	12 patients	L-shaped titanium mesh	-
Atef M [6]	*Clin Implant Dent Relat Res*	2020	Titanium mesh	RCT	20 patients	Titanium mesh	Collagen membrane
Hartmann A [80]	*BMC Oral Health*	2020	Customized titanium mesh	Retrospective study	98 patients	Titanium mesh	
Maiorana C [81]	*Materials (Basel, Switzerland)*	2020	Titanium meshes	Pilot study	8 patients	Peri-implant defects treated with titanium mesh	-
Malik R [82]	*J Maxillofac Oral Surg*	2020	Titanium mesh	Non-RCT	16 patients	Titanium mesh	-
Tallarico M [83]	*Materials (Basel, Switzerland)*	2020	3D titanium meshes	Case report	1 patient	3D titanium meshes	-
Li L [84]	*Clin Implant Dent Relat Res*	2021	3D titanium meshes	Retrospective study	16 patients	3D titanium meshes	-
Chiapasco M [85]	*Clin Oral Implants Res*	2021	CAD/CAM titanium mesh	Retrospective study	41 patients	CAD/CAM titanium mesh	-
Cucchi A [86]	*Int J Oral Implantology*	2021	Titanium mesh	RCT	40 patients	Titanium mesh	PTFE-reinforced membrane
Cucchi A [87]	*Clin Oral Implants Res*	2021	Custom-made CAD/CAM titanium meshes	RCT	30 patients	Titanium mesh with resorbable membranes	Titanium mesh without resorbable membranes
De Santis D [88]	*Medicina (Kaunas)*	2021	Digital Customized Titanium Mesh	Case series	5 patients	Custom-made CAD/CAM titanium meshes	-
Dellavia C [89]	*Clin Implant Dent Relat Res*	2021	Custom-made CAD/CAM titanium meshes	Cohort study	20 patients	Custom-made CAD/CAM titanium meshes	-
Kadkhodazadeh M [90]	*Oral Maxillofac Surg*	2021	Titanium meshes	Pilot study	7 patients	Titanium mesh with resorbable membranes	-
Lee SR [13]	*Materials (Basel, Switzerland)*	2021	3D titanium meshes	RCT	28 patients	Titanium mesh with cross-linked collagen membrane	Titanium mesh with non-cross-linked collagen membrane
Li S [91]	*Int J Oral Sci*	2021	Digital titanium mesh	Non-RCT	40 patients	Digital titanium mesh	Resorbable membranes
Maiorana C [92]	*J Contemp Dent Pract*	2021	Titanium meshes	Non-RCT; split-mouth study	5 patients	Titanium mesh	Dense polytetrafluoroethylene membrane
Wang X [93]	*Biomed Res Int*	2021	Titanium mesh membranes CGF membranes	Non-RCT	18 patients	Titanium mesh membranes and CGF membranes	-
Yoon JH [94]	*Maxillofac Plast Reconstr Surg*	2021	Titanium mesh	Case report	1 patient	Titanium mesh and pedicled buccal fat pad	-
Bertran Faus A [95]	*Materials (Basel)*	2022	Custom-made CAD/CAM titanium meshes	Case report	1 patient	Custom-made titanium mesh	-
Boogaard MJ [20]	*Compend Contin Educ Dent*	2022	Custom-made CAD/CAM titanium meshes	Case series	2 patients	Custom-made titanium mesh	-
Del Barrio RAL [96]	*J Oral Implantol*	2022	Titanium mesh frame (TF)	Case report	1 patient	Titanium mesh combined with recombinant human bone morphogenetic protein-2, deproteinized bovine bone mineral	
Gelețu GL [97]	*Medicina (Kaunas)*	2022	Custom-made CAD/CAM titanium meshes	Case report	1 patient	Custom-made titanium mesh	-
Hartmann A [98]	*Clin Oral Implants Res*	2022	Titanium mesh frame (TF)	Non-RCT	21 patients	Bone regeneration (GBR)/titanium mesh (TM)	-
Levine RA [99]	*Int J Periodontics Restorative Dent*	2022	Titanium mesh frame (TF)	Retrospective study	48 patients	Ti mesh guided bone regeneration	-
Lim J [100]	*Materials (Basel)*	2022	Titanium mesh frame (TF)	RCT	18 patients	Inorganic bovine bone materials (Bio-Oss)	A-Oss xenograft (Osstem, Seoul, Korea),
Majewski P [101]	*Int J Periodontics Restorative Dent*	2022	Titanium mesh frame (TF)	Case series	6 patients	Bone regeneration (GBR)/titanium mesh (TM)	-
Müller J [102]	*Case Rep Dent*	2022	Titanium mesh frame (TF)	Case report	1 patient	CAD CAM Ti mesh guided bone regeneration with previous bisphosponate treatment	-
Poomprakobsri K [103]	*J Oral Implantol*	2022	Titanium mesh frame (TF)/fixation screws	Retrospective study	-	Group 1: resorbable barrier Group 2: non-resorbable barrier Group 3: titanium-mesh barrier	-
Yang W [104]	*BMC Oral Health*	2022	Titanium mesh frame (TF)	non-RCT	20 patients	Defect volume < 150 mm^2^	Defect volume > 150 mm^2^
Abaza AWAAB [3]	*Int J Oral Maxillofac Implants*	2023	Titanium mesh frame (TF)	RCT	38 patients	Bone regeneration (GBR)/3D-printed individualized titanium mesh (3D-PITM)	Collagen group
Attia R [105]	*Int J Periodontics Restorative Dent*	2023	Titanium mesh frame (TF)	RCT	14 patients	Bone regeneration/coronally advanced lingual flap (CALF)	Bone regeneration/no coronally advanced lingual flap (CALF)
Bahaa S [106]	*Int J Oral Maxillofac Surg*	2023	Titanium mesh frame (TF)	RCT	40 patients	Flap groups: -Incision (PRI) -Double flap incision (DFI) -Modified periosteal releasing incision (MPRI) -Coronally advanced lingual flap (CALF)	-
Chen D [21]	*Clin Implant Dent Relat Res*	2023	CAD/CAM titanium mesh frame (TF)	Retrospective study	30 patients	Screw-position-guided template	No screws for position-guided template
Kurtiş B [22]	*J Oral Implantol*	2023	CAD/CAM titanium mesh frame (TF)	Case report	1 patient	Vertical bone augmentation/titanium mesh	-
Nan X [107]	*Clin Oral Implants Res*	2023	CAD/CAM titanium mesh frame (TF)	Retrospective study	59 patients	Bone regeneration (GBR)/3D-printed individualized titanium mesh (3D-PITM)	-
Onică N	*Healthcare (Basel)*	2023	CAD/CAM titanium mesh frame (TF)	Case report	1 patient	Bone regeneration (GBR)/3D-printed individualized titanium mesh (3D-PITM)	-
Onodera K [108]	*J Clin Med*	2023	Titanium mesh frame (TF) for severe mandibular bone defects	Retrospective study	18 patients	Custom-made titanium mesh	-
Songhang Li [109]	*Clin Implant Dent Relat Res*	2023	Titanium mesh frame (TF)	Retrospective study	36 patients	Titanium mesh stabilized with resorbable sutures	Titanium mesh stabilized with titanium screws
Wen SC [110]	*Int J Periodontics Restorative Dent*	2023	Titanium mesh frame (TF)	Case series	3 patients	Pre-trimmed TFs/graft and membrane	Pre-trimmed TFs/graft, no collagen membrane
Zhang G [111]	*J Esthet Restor Dent*	2023	Titanium mesh frame (TF)	Case series	3 patients	Tooth-supported TFs	-

**Table 4 dentistry-13-00052-t004:** Studies included after the literature screening. The synthesis was performed considering the technique, complications, bone graft study outcome, findings, and follow-up.

First Author	Journal	Year	Technique	Complications	Biomaterial/Membrane Type	Outcome	Follow-Up
Boyne PJ [24]	*Head Neck Surg*	1983	Titanium mesh	Patient #1: additional graft due to insufficient gain	-	Regenerated residual mandible segments from hemi-mandibulectomy	8–12 years
L Malchiodi [25]	*Int J Oral & Maxillofacial Imp*	1998	Titanium mesh	Patient #1: dehiscences around 3 implants	Autogenous bone graft	Higher width for alveolar ridge (mean: 5.65 mm; range: 5.20–6.10 mm)	8 months
von Arx T [26]	*Int J Periodontics Restorative Dent*	1998	Titanium mesh	None	Autogenous bone graft	All sites successfully treated	-
von Arx T [27]	*Clin Oral Implants Res*	1999	Titanium mesh	Implant complications (8): exposure (~6.5 mm), dehiscences (80%), fenestrations (20%), mesh exposure (rate: 5%)	Autogenous bone graft	Mean vertical bone height: 5.8 mm Mean bone defect filling: 93.5%	6.6 months
Sumi Y [28]	*Oral Surg Oral Med Oral Pathol Oral Radiol Endod*	2000	Titanium mesh	None	Autogenous bone graft	Regenerated alveolar crest width: 6.5 mm to 8 mm Mean gain in crest width: 3.5 mm	6 to 9 months
Klug CN [29]	*J Oral Maxillofac Surg*	2001	Titanium mesh L-shaped osteotomy	Patient #1: distractor fracture Patient #2: dehiscence	-	Mean bone height: 7.5 mm	19 months
Maiorana C [30]	*Int J Oral Maxillofac Implants*	2001	Titanium mesh	None	Autogenous cancellous bone/Bio-Oss in 1:1 ratio	Augmentation procedure showed bone regeneration and the presence of vessels, indicating bone vitality	5 to 6 months
Artzi Z [31]	*Int J Oral Maxillofac Implants*	2003	Titanium mesh	None	Xenograft	Defect height: Before = 6.4 +/− 1.17 mm After = 1.2 mm +/− 0.63 Bone height: 5.2 +/− 0.79 mm Average bone fill: 81.2% +/− 7.98	9 months
Degidi M [32]	*J Oral Implantol*	2003	Titanium mesh	None	Autogenous bone graft with bone-resorbable membrane	All cases had a good esthetic result after the restorative procedures	7 years
Proussaefs P [33]	*Int J Periodontics Restorative Dent*	2003	Titanium mesh	Mesh exposure with no compromise of graft	Bio-Oss, autogenous bone graft	Ridge augmentation: 2.86 mm (vertical), 3.71 mm (horizontal) Bone grafted area: 36.4% Graft resorption: 15.08%	6 months
Roccuzzo M [34]	*Clin Oral Implants Res*	2004	Titanium mesh	None	Autogenous bone graft, particulate xenograft	Mean vertical bone augmentation: 4.8 mm (range: 4–7 mm)	4.6 months
Proussaefs P [35]	*J Oral Implantol*	2006	Titanium mesh	Patient #2: early mesh exposure (2 weeks) Patient #4: latemesh exposure (>3 months)	Autogenous bone graft and Bio-Oss in 50:50 ratio	36.47% new bone formation 15.11% resorption	12 months
Kfir E [36]	*J Oral Implantol*	2007	Titanium mesh	Patient #7: early membrane exposure (47%)	Autologous platelet-rich fibrin	Sufficient bone augmentation in 8P	18 weeks
Longoni S [37]	*Int J Oral & Maxillofacial Imp*	2007	Titanium mesh	None	Regenaform demineralized freeze-dried bone allograft	-	18 months
Roccuzzo M [38]	*Clin Oral Implants Res*	2007	Titanium mesh	None	Autogenous onlay bone graft	Vertical augmentation: 5 mm T.g., 3.4 mm T.c. bone resorption: 13.5% T.g., 34.5% T.c.	4.6 months
Aikawa T [39]	*Oral Surg Oral MedOral Pathol Oral Radiol Endod*	2008	Ti mesh plate distraction device with titanium microscrews	None	None	4 mm widening at the first molar region Wider alveolar ridge	6 months
Pieri F [40]	*J Periodontol*	2008	Titanium mesh	19 micro-meshes exposed after 2 months (5.3%) 3 implants removed, bone resorption > 2 mm	70:30 mixture of autogenous bone graft and BBM (bovine bone mineral)	Mean vertical augmentation 3.71–1.24 mm Mean horizontal augmentation 4.16–0.59 mm	2 years
Corinaldesi G [41]	*Int J Oral Maxillofac Implants*	2009	Titanium mesh	4 micro-meshes exposed and removed (complication rate: 14.8%)	Autogenous bone graft	Vertical bone augmentation: 5.4 +/− 1.81 mm T. group: 4.5 +/− 1.16 mm C. group: implant CSR: 96.4%	3–8 years
Torres J [42]	*J Clin Periodontol*	2010	Titanium mesh	Control group: 28.5% mesh exposure Test group: none	Inorganic bovine bone (ABB), platelet-rich plasma (PRP)	Bone augmentation greater in test group: 97.3% in control group, 100% in test group	24 months
Ciocca L [43]	*Med Biol Eng Comput*	2011	Titanium mesh	-	Particulate autogenous bone graft, bovine demineralized bone	Mean vertical height difference in crestal bone: 2.57 mm Mean buccal–palatal increase: 3.41 mm	8 months
Misch CM [44]	*Int J Periodontics Restorative Dent.*	2011	Titanium mesh	-	Recombinant human BMP 2 ÷ acellular collagen sponge (rhBMP-2 ÷ ACS)	All 10 implants integrated	6 months
Cicciù M [45]	*Open Dent J*	2012	Titanium mesh	-	Absorbable collagen sponge, rhBMP-2	Mandibular continuity was regained	9 months
Her S [46]	*J Oral Maxillofac Surg*	2012	Titanium mesh	Ti mesh exposure rate: 26%	Puros-Bio-Oss autogenous graft	All 69 implants placed in function, 100% success rate	6–24 months
Miyamoto I [47]	*Clin Implant Dent Relat Res*	2012	Titanium mesh	Mesh exposure, infection, total/partial bone resorption, temporary neurological disturbances 1 implant failure	Autogenous particulate bone graft or iliac cancellous bone marrow grafts	Gain (mm): HV (H 3.7 ± 2.0; V 5.4 ± 3.4); H (3.9 ± 1.9) S > bone augmentation (H, 5.7 ± 1.4; V 12.4 ± 3.1) HV >> bone resorption (*p* < 0.05)	9 years
Ciocca L [48]	*Comput Methods Biomech Biomed Engin*	2013	CAD/CAM titanium mesh	-	Particulate, autogenous, bovine demineralized bone	Bone necessary for implants regenerated, bone augmentation in right lingual region extended beyond planned augmentation, maximum deviation < 1.5 mm	6 months
Funato A [49]	*Int J Periodontics Restorative Dent*	2013	Titanium mesh	Patient #1: flap dehiscence during healing period Patient #1: collagen membrane exposed during delayed period	Resorbable collagen membrane (cover of Ti mesh) Autogenous bone graft, Inorganic bovine bone particles + * Recombinant human platelet-derived growth factor BB	Mean vertical height of augmented bone: 8.6 ± 4.0 mm	8.0 ± 1.4 months
Atef M [50]	*Clin Implant Dent Relat Res.*	2014	Ti micromesh in maxillary sinus	-	-	Residual ridge height: 3.6 ± 1.6 mm Ridge height at follow-up: 9.63 ± 1.47 mm Volume of native bone: 30.3% ± 9.1% Volume of new bone: 55.3% ± 11.4%	6 months
Butura CC [51]	*Int J Oral Maxillofac Implants*	2014	Titanium mesh	-	rhBMP-2—inorganic bovine bone	Defects successfully regenerated with no additional surgery prior to implant placement or prosthetic restoration 14 implants placed and restored	6 months
Jung GU [52]	*J Korean Assoc Oral Maxillofac Surg*	2014	Titanium mesh	None	Particulate intraoral autologous bone + freeze-dried bone allograft in 1:1 volume ratio	New growth: 80% vital bone, 5% fibrous marrow tissue, 15% remaining allograft; all implants were functional	16 months
Katanec D [53]	*Coll Antropol*	2014	Titanium mesh	None	BMPs administered via an absorbable collagen sponge carrier (ACS) used for bone induction	Bone vertical gain: 5.5 mm (on the left), 5 mm (on the right), with 6 mm width Implant stability quotient (ISQ): 69 -75	24 months
Levine RA [54]	*Compend Contin Educ Dent*	2014	Titanium mesh	-	-	-	3 years
Lizio G [55]	*Int J Oral Maxillofac Implants*	2014	Titanium mesh	Mesh exposure occurred at 80% of augmented sites (0.73 cm^2^) at 2.17 months 16.3% LBV for every cm^2^ of mesh exposed	70:30 autogenous bone graft/inorganic bovine bone	LBV (lacking bone volume)-PBV (planned bone volume) Mean LBV (0.45 cm^3^) was 30.2% of the mean PBV (1.49 cm^3^)	8–9 months
Poli PP [56]	*Open Dent J*	2014	Titanium mesh	None	1:1 ratio autogenous bone graft mix with deproteinized inorganic bovine bone	Mean peri-implant bone loss of 1.743 mm on mesial side, 1.913 mm on distal side, from the top of implant head to the first visible bone–implant contact	88 months
Vrielinck L [57]	*J Craniofac Surg*	2014	Titanium mesh	No evidence of residual cyst	Xenograft	Restored mandibular shape and facial symmetry; promoted new bone formation to fill in the mandibular defects	5 years
De Angelis N [58]	*J Periodontics Restorative Dent*	2015	Titanium mesh	None	rhPDGF-BB, inorganic bovine bone particles, equine collagen sponge	Enough bone was regenerated to plan the implant placement according to the initial prosthetic plan and the patient requests	3 years
Di Stefano DA [59]	*J Contemp Dent Pract*	2015	Titanium mesh	None	30:70 mixture of autogenous bone graft and equine, enzyme-deantigenic collagen-preserved bone substitute	At follow-up, implants were perfectly functional, and the bone width was stable over time	24 months
Kim Y [60]	*Dent Traumatol*	2015	Titanium mesh	None	Xenograft + bone fragments from traumatic site Resorbable collagen membrane (on bone graft site)	Sufficiently preserved alveolar bone for implant placement	6 months
Lee JT [61]	*J Korean Assoc Oral Maxillofac Surg*	2015	Titanium mesh distraction osteogenesis		Xenograft	-	
Sumida T [62]	*J Craniomaxillofac Surg*	2015	Titanium mesh	Mucosal rupture (*p* = 0.27) in 1 patient with custom-made Ti mesh No severe infection (7.7%), 3 infections in control group (23.1%)	None	Operation time (min) t. group: 75.4 ± 11.6 c. group: 111.9 ± 17.8	6 months
Misch CM [63]	*Int J Oral Maxillofac Implants*	2015	Titanium mesh	None	30:70 ratio of autogenous bone graft and equine, enzyme- deantigenic collagen-preserving bone substitute	Implants perfectly functional, bone width stable over time, heterologous biomaterial were biocompatible and undergoing advanced remodeling and replacement with newly formed bone	24 months
Knöfler W [64]	*Int J Implant Dent*	2016	Titanium mesh	None	Graft materials (58.2%), membranes (36.6%), deproteinized bovine bone mineral (53%), autogenous bone particles (32.5%), native collagen membrane (74%)	Survival: 95.5%, significantly in augmented sites (*p* = 0.0025); best results for bone condensing method followed by lateral augmentation	20.2 years
Zita Gomes R [65]	*Biomed Res Int*	2016	Titanium mesh	Edema (48%), discomfort (40%), Ti mesh exposure (24%), graft loss in 2 cases (partial and complete + 1 failure)	Bio-Oss	Horizontal bone gain: 3.67 mm (±0.89) ISR: 97.5% Peri-implant marginal bone loss: 0.43 mm (±0.15)	1 year
Ahmed M [66]	*Int J Oral Maxillofac Surg*	2017	Titanium mesh	None	-	Evidence of new bone formation in both groups	6 months
Cucchi A [67]	*Clin Implant Dent Relat Res*	2017	Titanium mesh		-	Both GBR approaches: similar results regarding complications, vertical bone gain, and implant stability	3 years
Jegham H [68]	*J Stomatol Oral Maxillofac Surg*	2017	Titanium mesh	Mesh exposure visible with a circular flap dehiscence at follow-up (did not affect the successful regenerative outcomes)	Autogenous bone graft mixed with a xenograft	After mesh removal from grafted defects, space was completely filled with new hard tissue covered by a thin layer of soft tissue	4 months
Scarano A [69]	*Oral Implantol (Rome)*	2017	Titanium mesh	Mesh exposure in 1 of 3 P	Bone chips with resorbable membrane	Significant increase in alveolar width or height No residual bone defect observed	12.5 years
Alagl AS [70]	*J Int Med Res*	2018	Titanium mesh	None	Alloplast material mixed with a nano-bone graft	Newly formed ridge dimensions: 6 H and 10 mm V (original defect: 9 mm V) Complete filling of the defect, implant success	12 months
Ciocca L [71]	*J Oral Implantol*	2018	Titanium mesh	Mesh premature exposure (within 4 to 6 weeks) in 3 cases Delayed exposure (after 4 to 6 weeks) in 3 other cases Morbidity of mesh exposure (66%)	Particulate bone grafts, autogenous bone graft and inorganic bovine bone in 1:1 ratio	Mean mandibular bone augmentation: 3.83 mm Maxillabone augmentation: 3.95 mm	6–8 months
Inoue K [72]	*Implant dentistry*	2018	Titanium mesh	None	Xenograft (Bio-Oss)	-	6 months
Lorenz J [73]	*J Oral Implantol*	2018	Titanium mesh	-	Xenogeneic bone substitute (BO) with platelet-rich fibrin (PRF)	Complete rehabilitation and restoration of the patient’s oral function	16 months
Zhou M [74]	*J Oral Implantol*	2018	Titanium mesh	Infected graft in anterior mandible (tissue dehiscence); dehiscence 14 days after bone augmentation	Biocoral autologous bone	Total horizontal bone gain was 4.2 ± 0.5 mm	3 years
Cucchi A [75]	*J Oral Implantol*	2019	Titanium mesh	-	50:50 mixture of autogenous bone graft and xenograft	-	12 months
Di Stefano DA [59]	*Dent J (Basel)*	2019	Titanium mesh	-	Mixture of autogenous bone graft and equine-derived bone	Graft allowed effective bone formation (newly formed bone, residual Biomaterial and medullar spaces: 39%, 10%, and 51% of core volume)	6.5 years
Hartmann A [76]	*Implant Dent*	2019	Titanium mesh	Exposure (26)	Advanced and injectable platelet-rich fibrin (A- and I-PRF), resorbable membranes, autogenous bone graft, and Bio-Oss	-	12 months
Mounir M [77]	*Clin Implant Dent Relat Res*	2019	Titanium mesh				
Tallarico M [78]	*Materials (Basel)*	2019	Titanium mesh	Exposure after 1 month (1)	Xenograft/platelet-rich fibrin (PRF)	The mean bone gain was 5.06 ± 1.13 mm	18 months
Zhang T [79]	*Clin Implant Dent Relat Res*	2019	Titanium mesh	-	Xenograft	Average bone gain values were 3.61 ± 1.50 mm vertically and 3.10 ± 2.06 mm horizontally	41 months
Atef M [6]	*Clin Implant Dent Relat Res*	2020	Titanium mesh	Exposure (3)	1:1 mixture of autogenous and inorganic bovine bone	Mean bone gain of 4.0 mm for collagen group and 3.7 mm for titanium mesh group	6 months
Hartmann A [80]	*BMC Oral Health*	2020	Titanium mesh	Exposure (17)	Xenograft/A^®^-PRF	-	6 months
Maiorana C [81]	*Materials (Basel, Switzerland)*	2020	Titanium mesh for peri-implant defects	-	Xenograft	A mean horizontal bone gain of 4.95 ± 0.96 mm, and a mean horizontal thickness of the buccal plate of 3.25 ± 0.46 mm	8 months
Malik R [82]	*J Maxillofac Oral Surg*	2020	Titanium mesh	-	Novabone Putty	The mean vertical height of augmented bone was 4.825 ± 1.1387 mm	12 months
Tallarico M [83]	*Materials (Basel, Switzerland)*	2020	Titanium mesh	-	Xenograft	Implant successfully supported rehabilitation; no exposure or infection were documented	12 months
Li L [84]	*Clin Implant Dent Relat Res*	2021	Titanium mesh	-	Xenograft	Mean gain: 636.20 ± 341.18 mm^3^	9 months
Chiapasco M [85]	*Clin Oral Implants Res*	2021	Titanium mesh	Exposure (11)	Autogenous bone chips and bovine bone mineral (BBM).	The mean vertical and horizontal bone gains after reconstruction were 4.78 ± 1.88 mm and 6.35 ± 2.10 mm	18 months
Cucchi A [86]	*Int J Oral Implantology*	2021	Titanium mesh	-	50:50 bone mixtures of allograft	BV/TV, MatV/TV, and StV/TV in regenerated bone were 28.8%, 8.9%, and 62.4%, respectively In group B, the values of BV/TV, MatV/TV, and StV/TV were 30.0%, 11.0%, and 59.0%	9 months
Cucchi A [87]	*Clin Oral Implants Res*	2021	Titanium mesh	Exposure (test: 2; control 4) Implant failure (3)	50:50 bone mixtures of allograft	Better results for Mesh+ group (13%) compared to group Mesh- (33%)	12 months
De Santis D [88]	*Medicina (Kaunas)*	2021	Titanium mesh	-	Xenograft	An average horizontal gain of 3.6 ± 0.8 mm and a vertical gain of 5.2 ± 1.1 mm	9 months
Dellavia C [89]	*Clin Implant Dent Relat Res*	2021	Titanium mesh	-	Autogenous bone graft and deproteinized bovine bone (1:1).	35.88% new lamellar bone, 16.42% woven bone, 10.88% of osteoid matrix, 14.10% of grafted remnants, and 22.72% of medullary spaces	9 months
Kadkhodazadeh M [90]	*Oral Maxillofac Surg*	2021	Titanium mesh	-	Autogenous bone, allogenic graft material, and acellular dermal matrix	The mean marginal bone loss and bone gain were 4.4 ± 1.2 mm and 2.9 ± 0.9 mm	60 months
Lee SR [13]	*Materials (Basel, Switzerland)*	2021	Titanium mesh	Exposure (test: 1; control: 1)	Xenograft (Bio-Oss)	The mean HG rate was 84.25% ± 14.19% in the CCM group and 82.56% ± 13.04% in the NCCM group	6 months
Li S [91]	*Int J Oral Sci*	2021	Titanium mesh	Exposure (titanium mesh: 10%)	Autogenous bone graft and xenograft (Bio-Oss)	The percentage of resorption after 6 months of healing with resorbable membrane coverage reached 37.5%; however, it was only 23.4% with the titanium mesh	12 months
Maiorana C [92]	*J Contemp Dent Pract*	2021	Titanium mesh	Exposure (2)	1:1 ratio of autogenous bone to deproteinized bovine bone	Mean vertical bone gain of 4.2 and 1.5 mm was achieved in d-PM and TM groups	8 months
Wang X [93]	*Biomed Res Int*	2021	Titanium mesh	Exposure (1)	Xenograft/CGF	The thickness of the labial bone was 3.01 mm (±0.23), 2.96 mm (±0.21), 2.93 mm (±0.19), and 2.92 mm (±0.16) at the time of the second surgery, and 6 months, 1 year, and 2 years after the surgery	24 months
Yoon JH [94]	*Maxillofac Plast Reconstr Surg*	2021	Titanium mesh	Fistula	Buccal fat pad	-	12 months
Bertran Faus A [95]	*Materials (Basel)*	2022	Titanium mesh	None	Autogenous bone graft and xenograft mix in 70:30 ratio	Width: 1.84 and 1.92 mm; height: 3.78 mm	6 months
Boogaard MJ [20]	*Compend Contin Educ Dent*	2022	Titanium mesh	Exposure	-	Case 1: vertical gain—4.1 mm, width gain—8.7 mm; Case 2: vertical gain—6.7 mm, width gain—10.8 mm	5.5 months
Del Barrio RAL [96]	*J Oral Implantol*	2022	Titanium mesh	None	Recombinant human bone morphogenetic protein-2, deproteinized bovine bone mineral	These techniques were shown to be effective after 3 years of follow-up	3 years
Gelețu GL [97]	*Medicina (Kaunas)*	2022	Titanium mesh	None	Allograft bone substitute granules	Length: 11.63 mm; height: 10.34 mm	6 months
Hartmann A [98]	*Clin Oral Implants Res*	2022	Titanium mesh	Pain, suppuration, BOP, implant bone resorption	Autogenous bone and xenograft (Bio-Oss)	MBL: 0.13 ± 1.84 mm (mesial); −0.13 ± 1.73 mm (distal)	5 years
Levine RA [99]	*Int J Periodontics Restorative Dent*	2022	Titanium mesh	22% minor exposure	Allograft, cellular allograft, bovine xenograft/recombinant human platelet-derived growth factor, autogenous platelet-rich growth factor, and recombinant human bone morphogenetic protein-2	Mean horizontal gain: 4.7 ± 1.6 mm	8 months
Lim J [100]	*Materials (Basel)*	2022	Titanium mesh	None	Inorganic bovine bone materials (Bio-Oss) and A-Oss xenograft (Osstem, Seoul, Korea),	Grafted volume: Bioss group—1.70 ± 0.50 cc; Bio-Oss group—1.94 ± 0.26 cc	1 year
Majewski P [101]	*Int J Periodontics Restorative Dent*	2022	Titanium mesh	50% minor exposure	Xenograft/collagen membrane	Horizontal gain: 2 mm; vertical gain: 2.75 mm	6 months
Müller J [102]	*Case Rep Dent*	2022	Titanium mesh	None	Xenograft (Bioss)	Volume gain: 1.3–1.4 cm^3^	6 months
Poomprakobsri K [103]	*J Oral Implantol*	2022	Titanium mesh	Cumulative exposure rate: 36.9% Resorbable barrier ER: 23.3% Titanium mesh ER: 68.9%; Non-resorbable ER: 72.7%	Xenograft	Grafted bone dimensional loss with barrier exposure (58.3%) and no barrier exposure (44.1%)	6 months
Yang W [104]	*BMC Oral Health*	2022	Titanium mesh	Exposure: 1 minor, 1 major	Autogenous bone graft/deproteinized bovine bone mineral/iPRF	Mean deviation from planned GBR: −0.26 ± 0.35 mm	-
Abaza AWAAB [3]	*Int J Oral Maxillofac Implants*	2023	Titanium mesh	Control: dehiscence and infection (1) Test: premature/delayed exposure (4)	Autogenous bone graft and inorganic bovine bone graft mix at 50:50 ratio	Bone width in Control group: 7.3 ± 0.9 mm; bone width in Test group: 7.0 ± 0.9 mm	6 months
Attia R [105]	*Int J Periodontics Restorative Dent*	2023	Titanium mesh	Test: no exposure Control: 83% exposure	100% Xenograft biomaterial	Lingual flap advancement in Control group: 3.9 ± 1.1 mm; Test group: 14.4 ± 3.8 mm Buccal flap advancement in Control group: 15.8 ± 2.1 mm; Test group: 10.5 ± 1.4 mm	9 months
Bahaa S [106]	*Int J Oral Maxillofac Surg*	2023	Titanium mesh	None	Alloplastic bone	CALF group: 4.12 ± 1.37 mm; PRI group: 2.60 ± 1.36 mm; DFI group: 3.88 ± 1.70 mm; MPRI group: 3.44 ± 1.30 mm	6 months
Chen D [21]	*Clin Implant Dent Relat Res*	2023	Titanium mesh		100% xenograft biomaterial	No significant difference in 3D surgical positioning between the two groups	-
Kurtiş B [22]	*J Oral Implantol*	2023	Titanium mesh	None	Autogenous bone graft, deproteinized bovine bone mineral, injectable platelet-rich fibrin/collagen membrane	Successfully implant supported rehabilitation; no exposure or infection were documented	6 months
Nan X [107]	*Clin Oral Implants Res*	2023	Titanium mesh	Exposure rate: 32.8%	100% xenograft biomaterial	Width: 5.22 ± 3.19 mm Height: 5.01 ± 2.83 mm Volume bone gain: 588.91 ± 361.23 mm^3^	6 months
Onică N	*Healthcare (Basel)*	2023	Titanium mesh	None	Xenograft allograft and an autograft/collagen membrane mix	Implant successfully supported rehabilitation; no exposure or infection were documented	6 months
Onodera K [108]	*J Clin Med*	2023	Titanium mesh/neoplasm resection	Chronic pus discharge Exposure	Autogenous particulate cancellous bone and marrow graft	Augmented length: 3.21 ± 4.94 (SD) mm Marginal bone augmentation length: −0.15 ± 0.37 mm Segmental defects length: 4.89 ± 5.34 mm	-
Songhang Li [109]	*Clin Implant Dent Relat Res*	2023	Titanium mesh	None	Deproteinized bovine bone mineral (Bio-Oss) mixed with autogenous bone	-	6 months
Wen SC [110]	*Int J Periodontics Restorative Dent*	2023	Titanium mesh	None	50%/50% mixture of autograft/bovine xenograft + collagen membrane	8.0 ± 1.0 mm horizontal bone gain 3.0 ± 0.0 mm vertical bone gain Histomorphometry: 42.8% new vital bone, 18.8% residual bone graft particles, 38.4% bone marrow	8 months
Zhang G [111]	*J Esthet Restor Dent*	2023	Titanium mesh	None	100% xenograft biomaterial	Vertical bone gain: 4.16 mm Horizontal gain: 7.48 mm	6 months

## Data Availability

All experimental data that support the findings of this study are available from the corresponding author upon request.

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
