# Peer review of "The Early Exposure Rate and Vertical Bone Gain of Titanium Mesh for Maxillary Bone Regeneration: A Systematic Review and Meta-Analysis"

_dentistry, 2025, doi:10.3390/dj13020052_

Round 1
Reviewer 1 Report
Comments and Suggestions for Authors
Thank you for submitting your paper to the journal. While the topic of bone augmentation in implant dentistry is highly relevant and significant, the paper has several major limitations that prevent it from meeting the quality standards required for publication in this journal. I have outlined these limitations and specific comments below for your reference.
Lack of Protocol Registration: There is no mention of prior registration of the study protocol in platforms such as PROSPERO. This absence impacts the research's transparency and raises questions about its methodology. For instance, why were case reports and retrospective studies included? Furthermore, why were mesh exposure and vertical bone gain specifically chosen as outcome measures? Were these outcomes pre-defined in a study protocol, or were they selected after reviewing the included articles based on available data? This lack of clarity introduces a significant source of bias in the review.
Unclear Outcome Measures: The outcome measures are not clearly defined. For example, are there specific criteria used to assess vertical bone gain? Regarding mesh exposure, did the authors differentiate between early and delayed exposure? Were other major complications, such as infection, considered? More detailed descriptions of outcome measures would strengthen the study.
Methodological Concerns: The authors state, "Due to guarantee a pairwise comparison, we selected studies considering a similar methodological conformity for further statistical calculations." This suggests an attempt to tailor the data to fit a meta-analysis, which could introduce bias and compromise the validity of the results.
Limited Details on Meta-Analysis: The methodology of the meta-analysis lacks sufficient detail. For example, what model (random or fixed effects) was used? While it appears from the forest plots that a fixed-effects model was employed, the authors provide no rationale for this choice. Were any sensitivity analyses or meta-regressions conducted? Additionally, the authors could have calculated the cumulative exposure rate for mesh membranes to provide more robust results.
Overstated Conclusions: The conclusions appear too strong given the limitations of the data presented in the study. The authors should consider revising the conclusions to better align with the quality and robustness of their findings.
Comments on the Quality of English LanguageNeeds revisions but is acceptable.
Author Response
RE: The Early Exposure Rate and Vertical Bone Gain of Titanium Mesh for Maxillary Bone Regeneration: the current evidence of a Systematic Review and Meta-Analysis.
Dear Editor,
Thank you very much for reviewing our manuscript. According to the guidelines for the manuscript processing, please find attached the revised version of the paper with the integrations highlighted in yellow in the tracked version submitted as supplemental material.
Review Report (Reviewer 1)
Thank you for submitting your paper to the journal. While the topic of bone augmentation in implant dentistry is highly relevant and significant, the paper has several major limitations that prevent it from meeting the quality standards required for publication in this journal. I have outlined these limitations and specific comments below for your reference.
Lack of Protocol Registration: There is no mention of prior registration of the study protocol in platforms such as PROSPERO. This absence impacts the research's transparency and raises questions about its methodology. For instance, why were case reports and retrospective studies included? Furthermore, why were mesh exposure and vertical bone gain specifically chosen as outcome measures? Were these outcomes pre-defined in a study protocol, or were they selected after reviewing the included articles based on available data? This lack of clarity introduces a significant source of bias in the review.
[ANSWER] Thank you for your precious comment. The prospero registration number of the protocol has been added:
The present review has been registered in PROSPERO database (CRD42024585970).
[ANSWER] Thank you for your precious comment.
According to the Cochrane review methodology, the present review should consider strategically a multiple search on electronic databases. In other hands, the difficulties in MesH terms indicators on this topic, the screening has been conducted considering all clinical studies applying no filters regarding the study design methodology for the full text evaluation, eligibility process and descriptive synthesis. Further statistical methods have been considered the applicability of sub-groups comparison. The main limit of this approach is indubitably a sensible decrease of the study data robustness and strenghtness that should be considered in relation to the findings emerged from the review process.
Unclear Outcome Measures: The outcome measures are not clearly defined. For example, are there specific criteria used to assess vertical bone gain? Regarding mesh exposure, did the authors differentiate between early and delayed exposure? Were other major complications, such as infection, considered? More detailed descriptions of outcome measures would strengthen the study.
[ANSWER] The following paragraphs has been added:
Data Analysis
A database was created using Excel software (Microsoft, Redmond,WA, USA) specifically for entering data collected from the included scientific product studies. The data collected were classified as follows according to the characteristics listed here: study design, sample size, regenerative technique, complication, biomaterial/resorbable membrane type, surgical flap technique and the follow up.
Outcome Measures
The outcome measures considered for the data analysis were the flap exposure occurred during the bone regeneration healing period (<6 months), vertical bone height and horizontal bone gain calculated at the follow-up using computed tomograpy assessment.
Methodological Concerns: The authors state, "Due to guarantee a pairwise comparison, we selected studies considering a similar methodological conformity for further statistical calculations." This suggests an attempt to tailor the data to fit a meta-analysis, which could introduce bias and compromise the validity of the results.
[ANSWER] Thank you for the precious indication. Our opinion is that on contrary, the homogeneity of the study methodology is necessary to improve the robustness of the meta-analysis. It is necessary for the review methodology to reduce the wide range of bias associated to several variables including: surgical technique, the procedure site (single/multiple edentulism), atrophy grading, mesh characteristics (including the porosity), adaptation technique, the using of stabilization screw, biomaterial used.
[ANSWER] The following statement has been added in the discussion section:
Our opinion is the homogeneity of the study methodology is necessary to improve the robustness of the meta-analysis. It is necessary for the review methodology to reduce the wide range of bias associated to several variables including: surgical technique, the procedure site (single/multiple edentulism), atrophy grading, mesh characteristics (including the porosity), adaptation technique, the using of stabilization screw, biomaterial used.
.
Limited Details on Meta-Analysis: The methodology of the meta-analysis lacks sufficient detail. For example, what model (random or fixed effects) was used? While it appears from the forest plots that a fixed-effects model was employed, the authors provide no rationale for this choice. Were any sensitivity analyses or meta-regressions conducted? Additionally, the authors could have calculated the cumulative exposure rate for mesh membranes to provide more robust results.
[ANSWER] Thank you for the precious indication. We perfectly recognize that a random effect model is more appropriated for the data analysis. The statistical methods and output have been revised. In the methods and results section. Also, the figures 5-7 has been corrected.
Overstated Conclusions: The conclusions appear too strong given the limitations of the data presented in the study. The authors should consider revising the conclusions to better align with the quality and robustness of their findings.
[ANSWER] The conclusion section has been revised:
Premising the weak robustness of the study data and the limitations of the present review, the strenghtness of the analytic findings no definitive assumptions could be determined and limited and worthy of further investigation in the literature. The research outcome seems to suggest that bone regeneration of more or less extensive defects using titanium mesh represents an excellent bone regeneration technique, which despite being performed with different methods by combining membranes and/or bone grafts of different types and with different combinations, despite possible complications, does not see the validity of the regenerative technique compromised. Within the critical points of the present investigation, no significant differences in bone exposure and vertical bone gain have been observed comparing the technique with membrane bone regeneration. The physical and morphological characteristics of the titanium mesh, which can also be customized therefore personalized following the conformation of the defect, guarantee the immobilization and stability of the defect and thus guide the regeneration and, when present, the optimal integration of the biomaterial. The management and the surgical passivity of the flaps seems to minimize the risk of exposure with a significant reduction of the complication incidence.

Reviewer 2 Report
Comments and Suggestions for Authors
Dear authors,
The work presented here is, with no doubt, coming from a wide investigation. However, the overall work is very difficult to follow, as the objectives are unclear, and the manuscript lacks structure and organization.
The whole 'discussion' part is a general discussion about titanium mesh, and does not refer to the huge survey work.
i.e Line 194-196 : "The use of Ti mesh in the reconstruction of localized bone defects in the edentulous maxilla demonstrates results of high predictability..." Is this sentence a general comment, or is it based on reliable data extracted from your review?
In addition, the meta-analysis part is hardly discussed. Therefore, we cannot get a clear and reliable message from the manuscript.
In our understanding, the role/objective of such systematic review and meta-analysis should be to lead to reliable conclusions, and not general discussions about the interest of titanium mesh.
Therefore, we recommend your manuscript to have major revisions, with deep reorganization.
Here are also side comments :
> Figure 1 : What is the meaning of the first box "Records excluded"
> In Figure legends, it is sometimes written Figure, sometimes Fig.
> Figure 3 : Why are there only 6 studies explored?
> Figure 6 and Figure 7 have the same legend
> Figure 3 and Figure 6 refers to the study from Bahaa et al., which is from 2023, not 2003
> Line 203-204 : What does this sentence mean?
Comments on the Quality of English LanguageSome sentences are missing verbs, or are too long with several complementary parts
Author Response
RE: The Early Exposure Rate and Vertical Bone Gain of Titanium Mesh for Maxillary Bone Regeneration: the current evidence of a Systematic Review and Meta-Analysis.
Dear Editor,
Thank you very much for reviewing our manuscript. According to the guidelines for the manuscript processing, please find attached the revised version of the paper with the integrations highlighted in yellow in the tracked version submitted as supplemental material.
Reviewer 2
Dear authors,
The work presented here is, with no doubt, coming from a wide investigation. However, the overall work is very difficult to follow, as the objectives are unclear, and the manuscript lacks structure and organization.
The whole 'discussion' part is a general discussion about titanium mesh, and does not refer to the huge survey work.
i.e Line 194-196 : "The use of Ti mesh in the reconstruction of localized bone defects in the edentulous maxilla demonstrates results of high predictability..." Is this sentence a general comment, or is it based on reliable data extracted from your review?
[ANSWER] The following sentence has been corrected:
The use of titanium mesh in the reconstruction of localized bone defects in the edentulous maxilla demonstrates results has been purposed with high predictability and safety of the method with postoperative course characterized by excellent healing and very low exposure and complication rate [113].
In addition, the meta-analysis part is hardly discussed. Therefore, we cannot get a clear and reliable message from the manuscript.
In our understanding, the role/objective of such systematic review and meta-analysis should be to lead to reliable conclusions, and not general discussions about the interest of titanium mesh.
[ANSWER] Thank you for your precious comment. The study methodology and rationale has been clarified:
According to the Cochrane review methodology, the present review should consider strategically a multiple search on electronic databases. In other hands, the difficulties in MesH terms indicators on this topic, the screening has been conducted considering all clinical studies applying no filters regarding the study design methodology for the full text evaluation, eligibility process and descriptive synthesis. Further statistical methods have been considered the applicability of sub-groups comparison. The main limit of this approach is indubitably a sensible decrease of the study data robustness and strenghtness that should be considered in relation to the findings emerged from the review process.
Therefore, we recommend your manuscript to have major revisions, with deep reorganization.
Here are also side comments :
> Figure 1 : What is the meaning of the first box "Records excluded"
[ANSWER] The figure 1 was made in accordance to the PRISMA guidelines. The first level of screening could consider here an intermediate step. In some cases, grey literature could be excluded in this phase.
> In Figure legends, it is sometimes written Figure, sometimes Fig.
> Figure 3 : Why are there only 6 studies explored?
[ANSWER] Thank you for your precious indication. The risk of bias assessment was performed for all studies, but here we presented only the output of the studies included for meta-analysis. If necessary we could include all data as supplemental material.
> Figure 6 and Figure 7 have the same legend
[ANSWER] The following caption has been corrected:
Fig. 7. Forest plot of comparison: 1 Exposure, outcome: 1.4 Vertical bone gain (VBG).
> Figure 3 and Figure 6 refers to the study from Bahaa et al., which is from 2023, not 2003
[ANSWER] The figures 3 and 6 has been corrected,
> Line 203-204 : What does this sentence mean?
[ANSWER] The following sentence has been corrected:
Titanium mesh has been indicated considered for a wide pool clinical occurrence including the peri-implant bone defects, maxillary atrophies, alveolar sockets and periodontal defects and other therapeutic applications [91].

Reviewer 3 Report
Comments and Suggestions for Authors
1) Abstract: is too generic and lacks specific numerical details on the results. Please, include key numerical values and a more decisive conclusion on the risks and benefits of the use of titanium mesh for maxillary bone regeneration
2) “In addition, also no-graft bone augmentation procedure has been purposed for sinus augmentation”. Please the authors to better explain this sentence (at line 40) in the context in which it is inserted.
3) Add some biomaterials and membranes history into introduction section and add citations into it.
4) Line 46: “For this scope, in literature it was described the application of non-46 resorbable membrane (i.e. polytetrafluoroethylene-PTFE) or resorbable membrane (i.e. 47 collagen).” Please explain characteristics of each one and the choice of using one rather than the other.
5) Line 49: please the authors specify better which category of membranes the Titanium mesh refers to.
6) Line 61: “In fact, the main source of titanium mesh failure is 61 determined by the early/late exposure of the mesh”. Early or late? please better explain the possible complications related to the use of titanium mesh
7) Line 95: Why are only membranes or resorbable biomaterials considered in the Data Analysis section?
8) Line 112: regularize the space after the stitch.
9) Line 133: please correct the word “autogeneus”
10) 3.2. Characteristics of the Studies Included: This paragraph has a completely disconnected and illogical discourse
11) Figure and Tables: PRISMA figure and some tables are redundant and unclear. Review the PRISMA figure for clarity and simplify the tables by retaining only the most relevant data.

There are grammatical errors and excessively long sentences. Perform a stylistic review to improve clarity and flow, reducing redundant or overly complex sentences.
Author Response
RE: The Early Exposure Rate and Vertical Bone Gain of Titanium Mesh for Maxillary Bone Regeneration: the current evidence of a Systematic Review and Meta-Analysis.
Dear Editor,
Thank you very much for reviewing our manuscript. According to the guidelines for the manuscript processing, please find attached the revised version of the paper with the integrations highlighted in yellow in the tracked version submitted as supplemental material.
Reviewer 3
1) Abstract: is too generic and lacks specific numerical details on the results. Please, include key numerical values and a more decisive conclusion on the risks and benefits of the use of titanium mesh for maxillary bone regeneration
[ANSWER] The following section has been corrected:
Abstract: Background: Titanium mesh bone regeneration represent a clinical procedure oriented to regenerate bone defects ensuring graft stability and biocompatibility. The aim of the present investigation was to evaluate the clinical effectiveness of titanium mesh procedure in terms of vertical bone gain and the exposure rate. Methods: The product screening and eligibility process has been assessed through Pubmed/MEDLINE, EMBASE and Google Scholars electronic databases by two operators. The articles selected has been classified considering the study design, regenerative technique, tested groups and materials, sample size, clinical findings and follow-up. The risk of bias calculation has been conducted on randomized controlled trials (RCTs) and non-randomized trials selected and a series of pairwise meta-analysis calculations has been assessed considering the vertical bone gain (VBG) and exposure rate. A significant difference in terms of lower exposure rate has been observed using coronally advanced lingual flap. (p<0.05). No difference has been observed considering the titanium mesh and GBR technique in terms of VBG (p>0.05) Results: The initial search output reported 288 articles while 164 papers have been excluded after the eligibility process. The descriptive synthesis considered a total of 97 papers and 6 articles were considered for pairwise comparison. Conclusions: Within the limits of the present investigation, titanium mesh procedure reported excellent VBG after the healing period. The mesh exposure rate seems to be drastically reduced considering a passive management of the surgical flap.
2) “In addition, also no-graft bone augmentation procedure has been purposed for sinus augmentation”. Please the authors to better explain this sentence (at line 40) in the context in which it is inserted.
[ANSWER] The following sentence has been removed:
In addition, also no-graft bone augmentation procedure has been purposed for sinus augmentation [12,13].
3) Add some biomaterials and membranes history into introduction section and add citations into it.
4) Line 46: “For this scope, in literature it was described the application of non-46 resorbable membrane (i.e. polytetrafluoroethylene-PTFE) or resorbable membrane (i.e. 47 collagen).” Please explain characteristics of each one and the choice of using one rather than the other.
[ANSWER] The following section has been added:
Hystorically, non-resorbable and titanium reinforced membranes has been purposed for guided bone regeneration procedure in the late 1980s in relation with the high mechanical stability and space maintaining [17]. The limitations of this technique is the necessity of second surgery for removal and the tendency to the exposure during the healing period [17].
5) Line 49: please the authors specify better which category of membranes the Titanium mesh refers to.
[ANSWER] Corrected
6) Line 61: “In fact, the main source of titanium mesh failure is 61 determined by the early/late exposure of the mesh”. Early or late? please better explain the possible complications related to the use of titanium mesh
[ANSWER] The following section has been corrected:
In fact, the main source of titanium mesh failure is determined by the early/late exposure of the mesh during the healing period, combined with the contamination of the bone graft that often irreversibly compromise the regenerative procedure[23].
7) Line 95: Why are only membranes or resorbable biomaterials considered in the Data Analysis section?
[ANSWER] The following section has been added:
According to the Cochrane review methodology, the present review should consider strategically a multiple search on electronic databases. In other hands, the difficulties in MesH terms indicators on this topic, the screening has been conducted considering all clinical studies applying no filters regarding the study design methodology for the full text evaluation, eligibility process and descriptive synthesis. Further statistical methods have been considered the applicability of sub-groups comparison. The main limit of this approach is indubitably a sensible decrease of the study data robustness and strenghtness that should be considered in relation to the findings emerged from the review process. Our opinion is the homogeneity of the study methodology is necessary to improve the robustness of the meta-analysis. It is necessary for the review methodology to reduce the wide range of bias associated to several variables including: surgical technique, the procedure site (single/multiple edentulism), atrophy grading, mesh characteristics (including the porosity), adaptation technique, the using of stabilization screw, biomaterial used. In fact, considering the wide range of biomaterials used and the differences in methodology. A meta-analysis was not possible considering the biomaterial grafts used for both of technique.
8) Line 112: regularize the space after the stitch.
[ANSWER] Corrected.
9) Line 133: please correct the word “autogeneus”
[ANSWER] Corrected.
10) 3.2. Characteristics of the Studies Included: This paragraph has a completely disconnected and illogical discourse
[ANSWER] The following section has been moved in paragraph 3.1.
In this systematic literature review were considered including retrospective case-control studies, prospective studies, cohort studies, case series and case reports, randomized controlled Trial, non-RCT, preliminary studies, comparative studies. (Table 2).
[ANSWER] The paragraph 3.2 has been corrected:
3.2. Characteristics of the Studies Included
An important discriminant of the studies included is the The descriptive synthesis reported that the most frequent graft used for bone regeneration were biomaterial used in the surgical technique autogenous bone and autogenous bone mixed with heterologous bone graft. Some studies differed in autogenous/heterologous mix ratio that ranged from 50:50 ratio to 70:30 mixture of autogenous bone and BBM (bovine bone mineral). On other hands, the combination of platelet-rich plasma (PRP), collagen sponge (rhBMP-2÷ACS), resorbable collagen membrane, alloplastic material mixed with a nano-bone graft has been also reported. The most frequent complication reported was the mesh exposure that has been correlated to a partial failure of the graft or in some cases a more extensive compromission of the bone graft. Other complications reported were infection, total/partial bone resorption, temporary neurological disturbances and implant failure. The follow up was heterogeneous considering the included studies that ranged from 5 months to 20.5 years.
11) Figure and Tables: PRISMA figure and some tables are redundant and unclear. Review the PRISMA figure for clarity and simplify the tables by retaining only the most relevant data.
[ANSWER] The figure 1 was made in accordance to the PRISMA guidelines. The first level of screening could consider here an intermediate step. In some cases, grey literature could be excluded in this phase.

Round 2
Reviewer 1 Report
Comments and Suggestions for Authors
I appreciate the authors' efforts to improve the quality of the study. However, due to the lack of an a priori protocol (with the PROSPERO registration submitted during or after review completion), the study does not meet the criteria for a 'systematic review' according to PRISMA guidelines. Therefore, I unfortunately maintain the view that the study is not suitable for publication in this journal.
Author Response
RE: The Early Exposure Rate and Vertical Bone Gain of Titanium Mesh for Maxillary Bone Regeneration: the current evidence of a Systematic Review and Meta-Analysis.
Dear Editor,
Thank you very much for reviewing our manuscript. According to the guidelines for the manuscript processing, please find attached the revised version of the paper with the integrations highlighted in yellow in the tracked version submitted as supplemental material.
Reply to Reviewer 1
I appreciate the authors' efforts to improve the quality of the study. However, due to the lack of an a priori protocol (with the PROSPERO registration submitted during or after review completion), the study does not meet the criteria for a 'systematic review' according to PRISMA guidelines. Therefore, I unfortunately maintain the view that the study is not suitable for publication in this journal.
[ANSWER] Thank you for the Reviewer recommendations but the author’s opinion is that the present investigation followed the PRISMA criteria and PROSPERO database registration. Please verify the following considerations in materials and methods section.
Screening of Scientific Products
The literature search was performed following to the criteria of the PICO guidelines (population, intervention, comparison, outcome), as shown in Table 1. The data collected from the systematic search were thus processed in accordance with the Preferred Reporting Items for Systematic Reviews and Meta-Analyses (PRISMA) guidelines. The present review has been registered in PROSPERO database (CRD42024585970). The Boolean search was carried out according to the strategy described in Table 2 and performed on the electronic databases PubMed, EMBASE, and Google Scholar (10/06/2024).
Reviewer 2 Report
Comments and Suggestions for Authors
Dear authors,
The work presented here is coming from a deep survey work. In the original version, we felt that very few was gained from this review and analysis work, which was frustrating because of the lack of reliable conclusion...
We appreciate the efforts made to clarify the methodology and discussion in the manuscript.
At that stage, the manuscript is acceptable for publication, to our point of view.
Author Response
RE: The Early Exposure Rate and Vertical Bone Gain of Titanium Mesh for Maxillary Bone Regeneration: the current evidence of a Systematic Review and Meta-Analysis.
Dear Editor,
Thank you very much for reviewing our manuscript. According to the guidelines for the manuscript processing, please find attached the revised version of the paper with the integrations highlighted in yellow in the tracked version submitted as supplemental material.
Reply to Reviewer 2
Dear authors,
The work presented here is coming from a deep survey work. In the original version, we felt that very few was gained from this review and analysis work, which was frustrating because of the lack of reliable conclusion...
We appreciate the efforts made to clarify the methodology and discussion in the manuscript.
At that stage, the manuscript is acceptable for publication, to our point of view.
[Answer] Thanks to the Reviewer 2 for the comment. The last version of the manuscript followed the following Academic Editor recommendations.
Reviewer 3 Report
Comments and Suggestions for Authors
Dear authors,
the suggestions have been implemented, the quality of the article has improved, both in terms of content and presentation.
Author Response
RE: The Early Exposure Rate and Vertical Bone Gain of Titanium Mesh for Maxillary Bone Regeneration: the current evidence of a Systematic Review and Meta-Analysis.
Dear Editor,
Thank you very much for reviewing our manuscript. According to the guidelines for the manuscript processing, please find attached the revised version of the paper with the integrations highlighted in yellow in the tracked version submitted as supplemental material.
Reply to Reviewer 3
Dear authors,
the suggestions have been implemented, the quality of the article has improved, both in terms of content and presentation.
[Answer] Thanks to the Reviewer 3 for the comment. The last version of the manuscript followed the following Academic Editor recommendations.